# Robo functions as an attractive cue for glial migration through SYG-1/Neph

**Zhongwei Qu[1], Albert Zhang[1], Dong Yan[1,2]\***

[1]Department of Molecular Genetics and Microbiology, Duke University Medical Center, Durham, United States; [2]Department of Neurobiology, Regeneration Next Initiative, Duke Center for Neurodegeneration and Neurotherapeutics, and Duke Institute for Brain Sciences, Duke University Medical Center, Durham, United States

**Abstract** As one of the most-studied receptors, Robo plays functions in many biological processes, and its functions highly depend on Slit, the ligand of Robo. Here we uncover a Slit-independent role of Robo in glial migration and show that neurons can release an extracellular fragment of Robo upon cleavage to attract glia during migration in *Caenorhabditis elegans*. Furthermore, we identified the conserved cell adhesion molecule SYG-1/Neph as a receptor for the cleaved extracellular Robo fragment to mediate glial migration and SYG-1/Neph functions through regulation of the WAVE complex. Our studies reveal a previously unknown Slit-independent function and regulatory mechanism of Robo and show that the cleaved extracellular fragment of Robo can function as a ligand for SYG-1/Neph to guide glial migration. As Robo, the cleaved region of Robo, and SYG-1/Neph are all highly conserved across the animal kingdom, our findings may present a conserved Slit-independent Robo mechanism during brain development.

## Introduction

The evolutionarily conserved Robo family was first identified in the classic genetic studies of *Drosophila* CNS axonal midline crossing (*Seeger et al., 1993*) and belongs to the immunoglobulin (Ig) superfamily. Besides their functions in axon guidance (*Dickson and Gilestro, 2006*), Robo is also involved in cell migration (*Killeen and Sybingco, 2008*; *Viveiros et al., 2011*; *Hinck, 2004*; *Grieshammer et al., 2004*), organogenesis (*Grieshammer et al., 2004*; *Mommersteeg et al., 2013*), cancer development (*Wang et al., 2008*), and immune cell regulation (*Wu et al., 2001*; *Branchfield et al., 2016*). In these processes Robo functions as the receptor for the guidance cue molecule – Slit. However, mutations of Robo cause abnormalities in *Caenorhabditis elegans* (*Zallen et al., 1998*; *Hao et al., 2001*), zebrafish (*Fricke et al., 2001*), and cultured mammalian neurons (*Hivert et al., 2002*; *Liu et al., 2004*) that have not been observed in Slit mutants, suggesting that there may be evolutionarily conserved Slit-independent aspects of Robo function and regulation.

Glial cells often need to migrate over long distances from their birthplace to appropriate regions where they form functional units with neurons or perform other roles (*Gilmour et al., 2002*; *Jarjour and Kennedy, 2004*; *Kinrade et al., 2001*; *Klämbt, 2009*). Some guidance molecules including those in the Netrin and Semaphorin families were shown to be involved in glial migration (*Kinrade et al., 2001*; *Jarjour et al., 2003*; *Sasse and Klämbt, 2016*; *Spassky et al., 2002*; *Unni et al., 2012*). In *Drosophila*, CNS-derived glial cells can move along nerves to reach their final position, and during migration the glial expression of ROBO2 receptor is required for preventing glial breakaway from the nerve in a Slit-dependent manner (*Sasse and Klämbt, 2016*).

Proteolytic modifications play important roles in regulating Ig superfamily proteins. One good example is the Netrin receptor DCC. DCC can be cleaved first at the extracellular region and subsequently at the intracellular domains (*Neuhaus-Follini and Bashaw, 2015*). The DCC intracellular

**\*For correspondence:**
dong.yan@duke.edu

**Competing interests:** The authors declare that no competing interests exist.

cleavage fragment can then translocate into the nucleus and equip DCC with transcriptional regulatory function (*Neuhaus-Follini and Bashaw, 2015*). Similarly, NCAM and EphrinA5 can go through extracellular cleavage to release adhesive interactions and cause cell detachment (*Brennaman et al., 2014*). *Drosophila* Robo can be cleaved at the FN3 (Fibronectin type-III) repeats of the extracellular region, and the extracellular cleavage fragment can recruit Son of sevenless (Sos) to mediate Slit-dependent midline repulsion (*Coleman et al., 2010*). However, in all these cases the cleaved fragments and the function of the receptors highly rely on their canonical ligands. It is still unknown whether the cleavage of Ig superfamily receptors can function in a fashion that is independent of their canonical ligands.

There are in total 56 glia in *C. elegans*, which include 50 neuroepithelial glia that ensheath sensory neuron receptive endings (*Oikonomou and Shaham, 2011*; *Shaham, 2015*). The largest sensory organ of the worm, called the amphid sensilla, is a pair of sensilla composed of 12 neurons and two glial cells each (*Ward et al., 1975*). These glia, called the amphid sheath (AMsh) and amphid socket (AMso) cells, form channels that ensheath the dendrites of sensory neurons in the amphid sensilla (*Oikonomou and Shaham, 2011*; *Ward et al., 1975*; *Perkins et al., 1986*). Furthermore, these glial cells are vital for the proper functioning of the neurons that they ensheath, where they can modulate neural activity through secreted molecules at the receptive endings and control neuron receptive ending shape (*Bacaj et al., 2008*; *Shaham, 2010*; *Singhvi et al., 2016*). In previous studies we show that the fate determination of *C. elegans* AMsh glial cells utilizes similar mechanisms as those in mammals (*Zhang et al., 2020a*). AMsh glia are born close to the nose of *C. elegans*, where their processes anchor to the nose region while the cell bodies migrate toward the nerve ring (*Shaham, 2015*; *Heiman and Shaham, 2009*), and the position of AMsh glia is important for their function (*Zhang et al., 2020b*). With all these advantages, AMsh glia have emerged as a powerful model to study the potential conserved mechanisms underlying glial development and function.

## SAX-3/Robo regulates glial migration in a Slit-independent manner

In an unbiased genetic screen for glial migration, we isolated two loss-of-function alleles of *sax-3*, the only Robo receptor in *C. elegans* (*Zallen et al., 1998*), with defects in AMsh glial migration. To quantify the migration of AMsh, we used the pharyngeal terminal bulb as a reference point and found that AMsh cell bodies reside at the side of and align with the center of pharyngeal terminal bulbs in day 1 (D1) young adults (*Figure 1a and b*). AMsh glia fail to migrate to their final positions in both *sax-3(lf)* alleles, and *yad10* animals display stronger phenotypes than *yad147* animals and show a similar extent of defects as those in the null allele of *sax-3 (ky123)* (*Zallen et al., 1998*), suggesting that *yad10* is likely a null allele of *sax-3* (*Figure 1a and b* and *Figure 1—figure supplement 1a and c*). To further characterize the *sax-3(lf)* phenotype we examined AMsh migration at different developmental stages in a single animal and found that *sax-3(lf)* animals exhibited migration defects in all post-embryonic stages, suggesting that *sax-3* is required for AMsh migration during development (*Figure 1—figure supplement 1b*).

Next, we examined the function of the canonical Robo ligand, Slit, in AMsh migration and found that loss-of-function of *slt-1(ok255*, null allele), the only Slit in *C. elegans*, did not cause any defects, suggesting that Robo functions in a Slit-independent manner to regulate AMsh migration (*Figure 1c*). As *sax-3* is involved in the regulation of nerve ring position (*Zallen et al., 1998*; *Sasakura et al., 2005*), it could be argued that the AMsh migration defect could originate from the anteriorly positioned nerve ring in *sax-3(lf)* mutants. To exclude this possibility, we examined the relative position of AMsh glial cells and the nerve ring. Results show that AMsh cell bodies reside posterior to the nerve ring in control animals but are misplaced anterior to the nerve ring in *sax-3(lf)* animals (*Figure 1—figure supplement 1d and e*), suggesting that the AMsh migration defect is not a side phenotype coming from a displaced nerve ring. As about 40% of *sax-3(lf)* animals display defects in head morphogenesis, called '*vab head*' (*Ghenea et al., 2005*), it is also possible that AMsh defects could be a result of abnormal head morphology. To rule out this possibility, we examined *sax-3(lf)* animals without '*vab head*' phenotypes and found that they displayed the same degree of AMsh defects as in all *sax-3(lf)* animals, supporting the conclusion that the AMsh migration defects in *sax-3(lf)* animals are independent of the '*vab head*' phenotypes (*Figure 1c*).

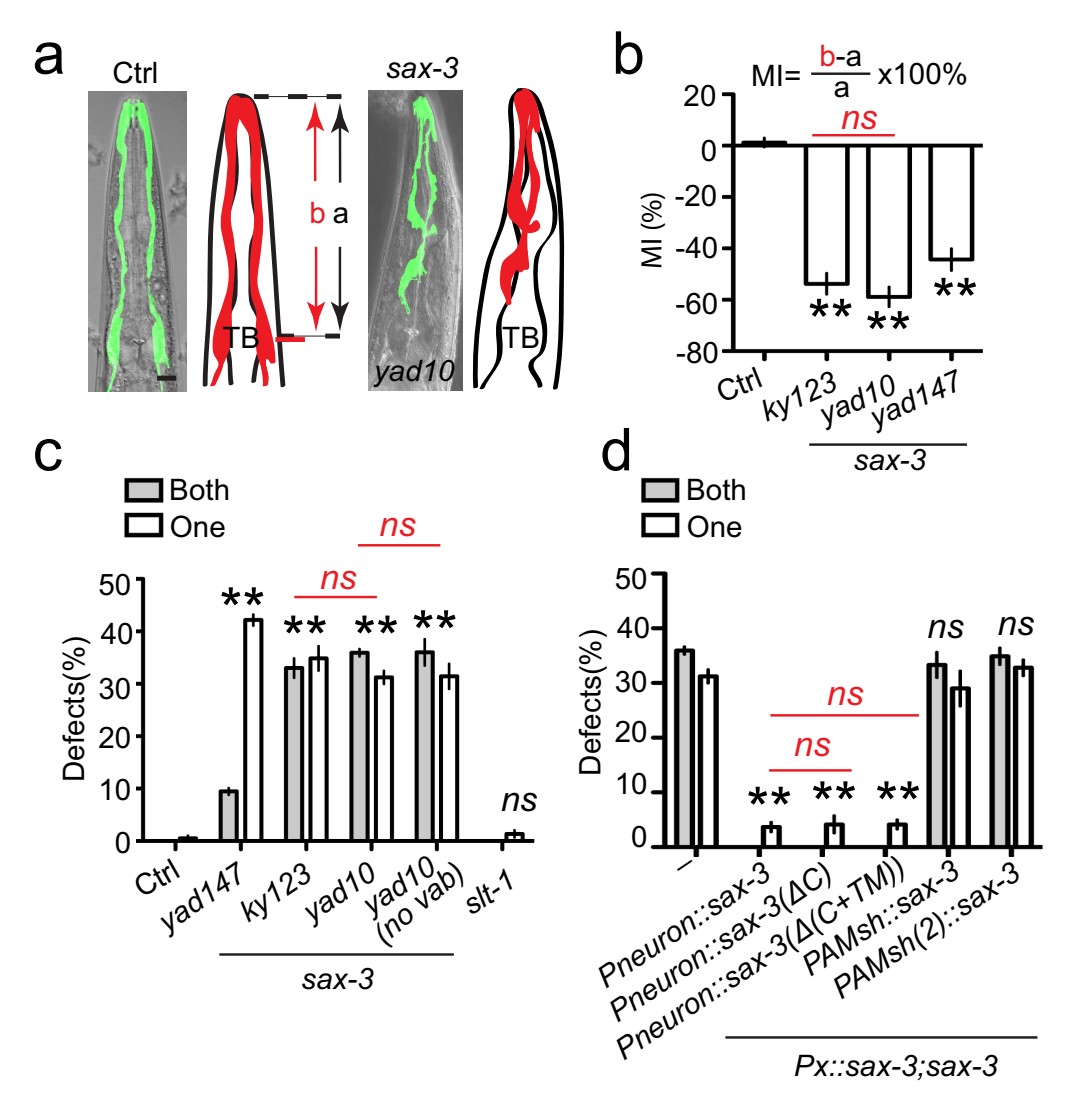

**Figure 1.** *sax-3*/Robo regulates AMsh glial migration in a Slit-independent fashion. (a) Loss-of-function in *sax-3* causes defects in AMsh glial migration. Confocal images and schematic representation of AMsh glia in control and *sax-3(yad10)* animals expressing *Pf53f4.13::GFP* (*yadIs48*). TB: pharyngeal terminal bulb. Scale bar, 10 μm. (b) Quantification of Migration Index (MI) in control and three alleles of *sax-3(lf)* animals. MI is calculated as (b–a)/a × 100%. As illustrated in the (a), 'a' represents the distance between the tip of nose and the center of the pharyngeal terminal bulb. 'b' shows the distance between the tip of nose and the center of the AMsh cell bodies. (c) The function of *sax-3* does not rely on *slt-1*. White and gray bars show the percentage of animals with defects in one AMsh or both AMsh glia respectively. '*no vab*': animals with normal head morphology. (d) Expression of a secreted SAX-3 extracellular fragment in neurons rescue AMsh defects in *sax-3(lf)* animals. Data shows the percentage of animals with AMsh migration defects. *Punc-33(Pneuron)*, *Pf53f4.13(PAMsh)*, and *Pf16f9.3(PAMsh(2))* are used as pan-neuronal and AMsh-specific promoters to express genes in neurons or AMsh glia. In (b), data are represented as mean ± SEM. One-way ANOVA test. **, $p<0.01$; Each point represents at least 30 worms. In (c) and (d), data are represented as mean ± SEM. Two-way ANOVA test. **, $p<0.01$; ns, no significant difference. Each point represents three experiments of at least 50 worms each.

The online version of this article includes the following figure supplement(s) for figure 1:

**Figure supplement 1.** Loss-of-function of *sax-3/*Robo causes AMsh migration defects.

## The extracellular cleavage fragment of SAX-3/Robo functions as an attractive cue for glial migration

To understand how *sax-3* regulates AMsh migration, we carried out tissue-specific rescue experiments. Unexpectedly, as a well-studied receptor, *sax-3* does not function cell-autonomously in AMsh migration, and instead expression of *sax-3* cDNA in neurons fully rescues *sax-3(lf)* AMsh phenotypes

(*Figure 1d*, *Figure 1—figure supplement 1g*). It is even more striking that expression of truncated forms of SAX-3 lacking the intracellular domains (ΔC) or both intracellular and transmembrane domains (Δ[C+TM]) fully rescues *sax-3* AMsh defects (*Figure 1d*, *Figure 1—figure supplement 1h*), while these transgenes did not affect the *sax-3(lf)* 'vab head' phenotypes (*Figure 1—figure supplement 1f and h*). A possible explanation for these observations is that a secreted extracellular fragment of SAX-3 may function as an attractive cue released by neurons to guide AMsh migration. To reach this possibility, either *sax-3* should have an alternative splicing form that only contains its extracellular region or SAX-3 protein should be cleaved in the extracellular region. The *C. elegans* genome has been well annotated and studied (*Gupta and Sternberg, 2003*), and *sax-3* does not have any known splicing forms containing only the extracellular region. Interestingly, *Drosophila* and human Robo have been shown to be cleaved at the FN3 repeats of the extracellular region (*Coleman et al., 2010*; *Seki et al., 2010*). As FN3 repeats are highly conserved between *C. elegans* SAX-3 and *Drosophila* Robo, we decided to test whether SAX-3 could also be cleaved in the extracellular region. We first generated transgenes expressing SAX-3::HA in the nervous system and collected animals at different developmental stages. As shown in *Figure 2a* and *Figure 2—figure supplement 1a*, we were able to detect a cleavage fragment around 70 kDa in different transgene lines and using different anti-HA antibodies. As the HA tag is fused to the C-terminus of SAX-3, based on the molecular weight (about 70 kDa) the cleavage site is likely to be in the FN3 repeats, which is consistent with what was found in *Drosophila* Robo (*Coleman et al., 2010*). More importantly, the extracellular cleavage happens from the embryonic 1.5-fold stage to early larval stages, when AMsh glia migrate from their birthplace to the nerve ring region (*Figure 2a* and *Figure 2—figure supplement 1a*). As the function of *sax-3* in AMsh migration is independent of Slit, we asked whether the cleavage of SAX-3 relied on Slit and found that the cleavage of SAX-3 was not regulated by Slit (*Figure 2b* and *Figure 2—figure supplement 1b*). To further confirm that the cleavage site is in the FN3 repeats, we generated transgenes expressing a mini SAX-3 extracellular region that only contains the FN3 repeats, transmembrane domain, and a GFP inserted between the signal peptide and the first FN3 repeat (GFP-FN3-TM) and examined the cleavage products of this fragment. We found that FN3 repeats were sufficient for the cleavage (*Figure 2c*), and based on the molecular weights of free GFP (27 kDa), GFP-FN3-TM (~80 kDa), and the cleavage product (~45 kDa) we concluded that the cleavage site was in the second repeat of FN3. We then generated transgenes expressing the GFP-FN3-TM fragment lacking one of the three FN3 repeats. As deletion of the first FN3 repeat (FN3-a) caused failure of expression, we were only able to examine the effect of removing the second (FN3-b) or the third FN3 (FN3-c) repeat. We found that the cleavage was abolished if the second FN3 repeat was absent, while deletion of the third FN3 repeat did not affect the cleavage (*Figure 2d*). Consistent with its important role in SAX-3 cleavage, extracellular cleavage was undetectable in transgenes expressing a truncated form of SAX-3 without the second FN3 repeat (*Figure 2e*). These data support the conclusion that the cleavage site of SAX-3 is in the second repeat of FN3. To test whether the extracellular cleavage of SAX-3 is essential for its function in AMsh migration, we examined the rescue ability of the truncated form of SAX-3 that lacks the second FN3 repeat and found that this mutant form of SAX-3 completely lost rescue ability (*Figure 2f*, *Figure 2—figure supplement 1c*). The extracellular cleavage of SAX-3 appears to show certain specificity for AMsh migration, as the same transgene (SAX-3 ΔFN3-b) fully rescued *sax-3(lf)* 'vab head' phenotypes (*Figure 1—figure supplement 1f*, *Figure 2—figure supplement 1c*). To further confirm the role of extracellular SAX-3 cleavage in AMsh migration, we generated a *sax-3(yad175)* deletion allele that lacks the second repeat of FN3 (FN3-b). *sax-3(yad175)* decreases *sax-3* mRNA to about 50% of that in control animals (*Figure 2—figure supplement 1d*) and appears to be a weak loss-of-function allele of *sax-3,* as both *sax-3(ky123,* presumptive null allele) and *sax-3(yad10)* show higher penetrance of *vab* head and AVM guidance defects (*Zallen et al., 1999*; *Figure 1—figure supplement 1f* and *Figure 2—figure supplement 1e*). Despite being a weak allele, *sax-3(yad175)* exhibits stronger AMsh migration defects when compared with any other *sax-3* alleles examined, supporting the critical role of SAX-3 extracellular cleavage in AMsh migration (*Figure 2f*). To illustrate the function of the extracellular cleavage fragment, we expressed a secreted SAX-3 extracellular fragment containing Ig 1–5 domains and the first FN3 repeat (FN3-a) under pan-neuronal (*Punc-33* and *Prgef-1*) or amphid neuron-specific (*Parl-13*) promoters, which mimics the cleavage product, and found that this fragment was able to fully rescue *sax-3(lf)* AMsh defects (*Figure 2f*, *Figure 2—figure supplement 1f*). Further experiments show that both Ig domains and the first FN3 repeat

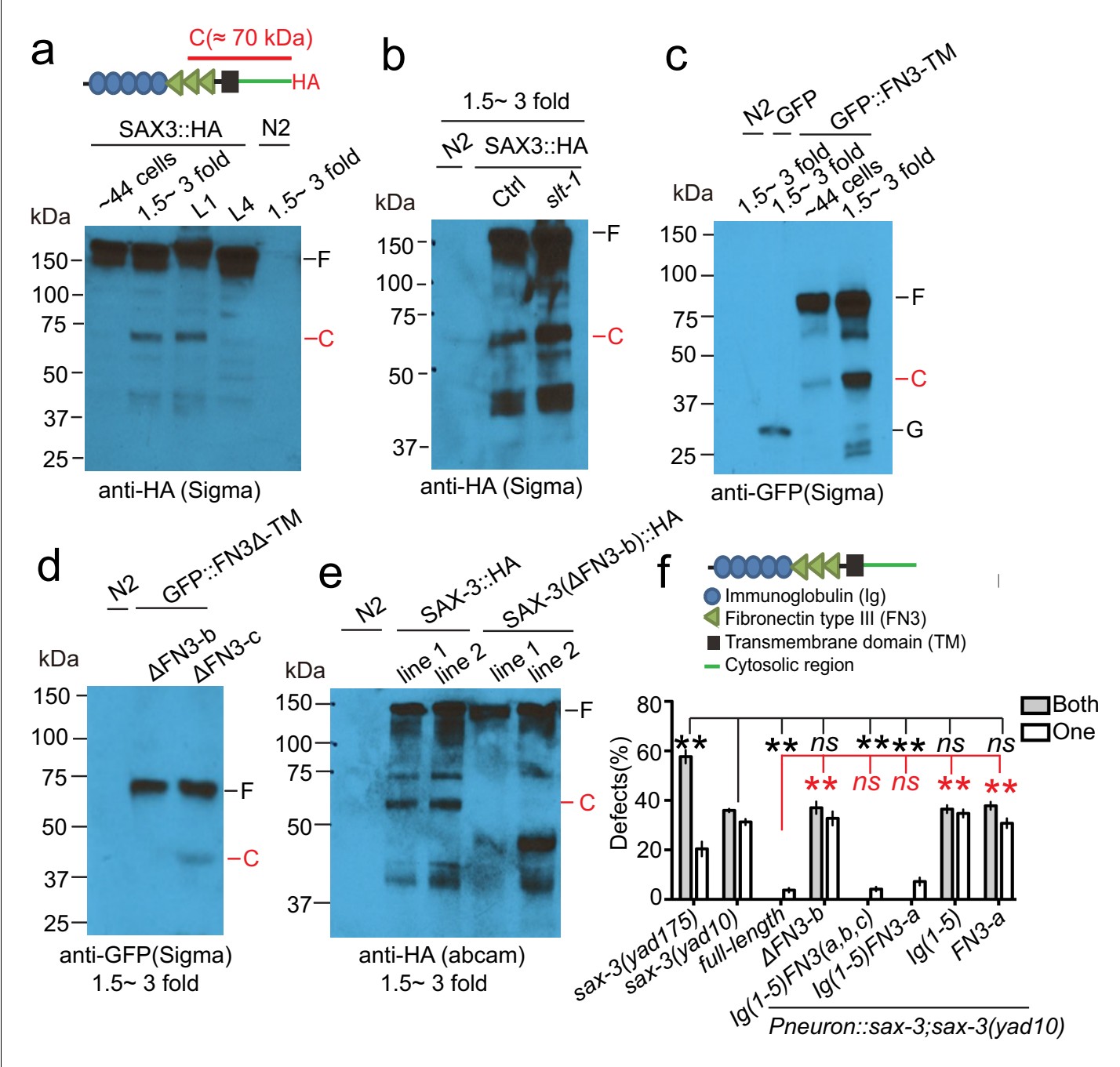

**Figure 2.** The extracellular cleavage of SAX-3 is required for AMsh glial migration. (**a**) SAX-3 is cleaved in the extracellular region during development. A schematic diagram at the top panel shows that the HA tag was fused at the C-terminal of SAX-3, and the leftover of the extracellular cleavage detected by the HA antibody is about 70 kDa. Results from western blot show a cleavage fragment of SAX-3 in later embryonic and early larval stages. F: full length SAX-3::HA; C: the C-terminal fragment generated by SAX-3 extracellular cleavage. (**b**) Loss-of-function of *slt-1*/Slit does not affect the cleavage of SAX-3. F: full length SAX-3::HA; C: SAX-3 cleavage fragment. (**c**) The FN3 repeats mediate SAX-3 cleavage. FN3: FN3 repeats; TM: transmembrane domain; F: full length; C: cleavage fragment; G: free GFP. (**d**) FN3-b (the second repeat of FN3 repeats) is required for the cleavage. F: full length; C: cleavage fragment. (**e**) Deletion of FN3-b prevents SAX-3 cleavage. Images show results from two independent transgenic lines with FN3-b deletion (ΔFN3-b). Although having slightly different expression patterns, they both lose the extracellular cleavage fragment of SAX-3. (**f**) Expression of a fragment mimicking the SAX-3 extracellular fragment fully rescues AMsh defects in *sax-3(lf)* animals. Data are represented as mean ± SEM. Two-way ANOVA test. **, $p < 0.01$; ns, no significant difference. Each point represents three experiments of at least 50 worms each. The online version of this article includes the following figure supplement(s) for figure 2:

**Figure supplement 1.** The Slit-independent SAX-3 cleavage.

(FN3-a) may be important for the function of the SAX-3 extracellular cleavage fragment in AMsh migration (*Figure 2f*).

## SYG-1/Neph is the receptor for the SAX-3/Robo extracellular cleavage fragment during glial migration

With the discovery of the function of the SAX-3 extracellular cleavage fragment in AMsh glial migration, we hypothesized that the cleavage fragment is released from neurons, where *sax-3* is expressed (*Zallen et al., 1998*), and functions as an attractive cue for glial migration. If that is the case, there should be a receptor in AMsh glia to interact with the SAX-3 cleavage fragment and mediate AMsh migration. To identify the potential receptor for the SAX-3 cleavage fragment, we generated a transgene expressing a functional FLAG::SAX-3 fusion protein, in which a FLAG tag was inserted between the signal peptide and the first Ig domain. We first confirmed that we could detect the developmental-stage-dependent SAX-3 cleavage in this transgene (*Figure 3a*). Then we carried out immunoprecipitation (IP) using anti-FLAG-coated beads on protein lysate collected from 44 cell and threefold stage embryos and submitted them for proteomics (*Figure 3—figure supplement 1a*). As the SAX-3 cleavage happens only in threefold but not in 44-cell stage embryos, we focused on membrane proteins that specifically interacted with SAX-3 in threefold stage embryos. Furthermore, Ig superfamily proteins have been shown to be able to interact with each other in many biological processes (*Cameron and McAllister, 2018*; *Wai Wong et al., 2012*), so we also focused our analysis on Ig superfamily proteins. *syg-1*, a nephrin family adhesion molecule (*Shen and Bargmann, 2003*), caught our attention (*Figure 3—figure supplement 1b*, *Supplementary file 2*) because it contains multiple Ig domains in its extracellular regions (*Zallen et al., 1998*; *Shen and Bargmann, 2003*). To confirm the interaction between SYG-1 and SAX-3, we first generated a transgene expressing SYG-1::HA in glia and FLAG::SAX-3 in neurons and then examined the interaction between SYG-1::HA and FLAG::SAX-3 by IP using anti-HA-coated beads. As shown in *Figure 3b*, the extracellular cleavage fragment but not the full-length SAX-3 can bind with SYG-1::HA, suggesting that SYG-1 can directly or indirectly bind with the SAX-3 extracellular cleavage fragment. With these findings, we further examined the function of *syg-1* in AMsh migration and found that *syg-1* was expressed in AMsh glia during migration and loss-of-function of *syg-1(ok3640, null allele)* caused similar AMsh migration defects as those in *sax-3(lf)* mutants (*Figure 3c–e*, *Figure 3—figure supplement 2a*). Furthermore, double mutants of *syg-1;sax-3* displayed the same extent of defects as in *sax-3* single mutants (*Figure 3d and e*), supporting the conclusion that *syg-1* and *sax-3* function in the same genetic pathway to regulate AMsh migration. As *syg-1* has been shown to function together with *syg-2*, another Ig super family protein, in synapse formation and axon branching (*Shen et al., 2004*; *Chia et al., 2014*), we examined the function of *syg-2* in AMsh migration and did not observe any abnormalities in *syg-2(lf)* mutants (*Figure 3e*). Consistent with its potential role as a receptor for the SAX-3 cleavage fragment, *syg-1* cell autonomously regulates AMsh migration (*Figure 3e*, *Figure 3—figure supplement 2b*). Further genetic analysis also showed that the conserved WIRS (WAVE Regulatory receptor sequence) in the SYG-1 C-terminal (*Chia et al., 2014*) is essential for its function in AMsh migration (*Figure 3e*), and interruption of WAVE complex function by mutating *gex-3*, one of the key components of the WAVE complex (*Chia et al., 2014*), cell autonomously caused similar AMsh migration defects as observed in *sax-3(lf)* and *syg-1(lf)* mutants (*Figure 3f,g*). Finally, to further test the ligand/receptor function of the SAX-3 cleavage fragment and SYG-1, we tested whether AMsh glia can further migrate along the body in transgenic animals that express the cleavage product of SAX-3 in the tail neurons (*Pitr-1b*) or tail hypodermis (*Plin-44*) of *sax-3(lf)* animals (*Ou et al., 2010*). We found that in these transgenes the migration distance of AMsh was almost three times that of control animals, and loss-of-function in *syg-1* completely blocked the over-migration phenotypes in these transgenes (*Figure 3h and i*). With this evidence we believe that the neuron-released SAX-3 extracellular cleavage fragment functions as an attractive cue, and SYG-1 functions as its receptor to mediate AMsh migration (*Figure 4*).

## Discussion

Studies in different organisms show that Robo functions as the receptor for the repulsive guidance cue Slit (*Dickson and Gilestro, 2006*), and loss-of-function in Robo causes many phenotypes that are shared by Slit mutants (*Dickson and Gilestro, 2006*). However, genetic studies also showed that

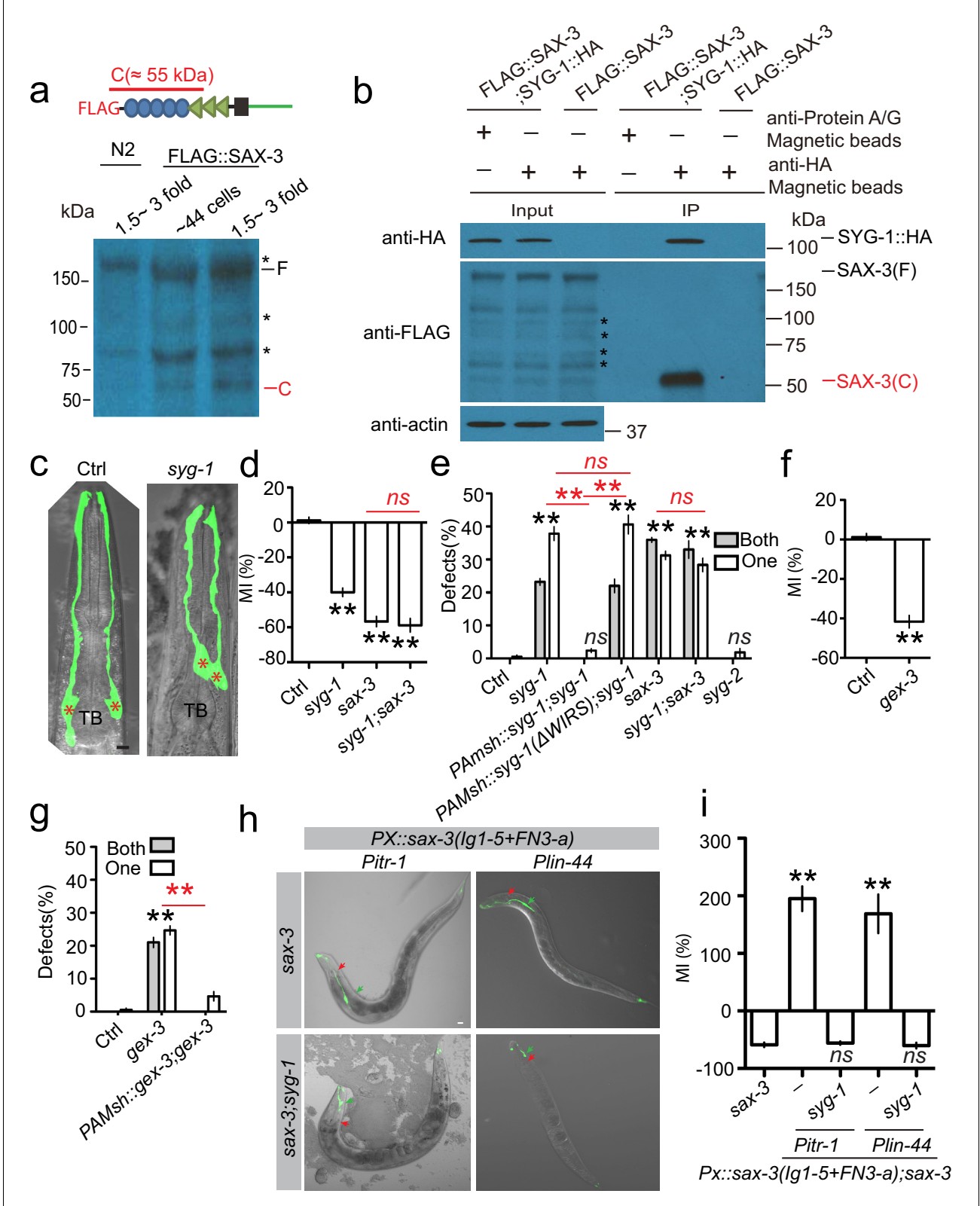

**Figure 3.** SYG-1 functions as a receptor for the SAX-3 extracellular cleavage fragment during AMsh glial migration. (a) Image from western blot show the SAX-3 extracellular fragment by N-terminal tagging FLAG transgenes. Stars (*) label bands that are nonspecifically recognized by the anti-FLAG antibody. A schematic diagram at the top panel shows that the FLAG tag was fused at the N-terminal of SAX-3, and the extracellular cleavage fragment detected by the FLAG antibody is about 55 kDa. (b) SYG-1 specifically binds with the SAX-3 extracellular cleavage fragment. SYG-1::HA expression is

*Figure 3 continued on next page*

*Figure 3 continued*

driven by the pan-glial promoter *Pmir-228*, and FLAG::SAX-3 is expressed under the pan-neuronal promoter *Punc-33*. Stars (*) label bands that are nonspecifically recognized by the anti-FLAG antibody. (c) Confocal images of AMsh glia in control and *syg-1(ok3640)* animals. TB: pharyngeal terminal bulb. Stars (*) label AMsh cell bodies. Scale bar, 10 µm. (d) Quantification of <u>M</u>igration <u>I</u>ndex (MI) in control, *syg-1*, *sax-3*, and *syg-1;sax-3* double mutants show that double mutants of *syg-1;sax-3* display similar defects as in *sax-3* single mutants. (e) *syg-1* cell autonomously regulates AMsh glial migration, and the WAVE regulatory receptor sequence is required for *syg-1* function. (f and g) *gex-3* cell autonomously regulates AMsh glial migration. Quantification of <u>M</u>igration <u>I</u>ndex (f) and the percentage of animals with AMsh defects (g) show that *gex-3* is required for AMsh migration. (h and i) Confocal images (h) and quantification (i) show that ectopic expression of a fragment mimicking the SAX-3 cleavage fragment in the tail neurons (*Pitr-1*) or hypodermis (*Plin-44*) causes *syg-1*-dependent over-migration of AMsh glia. Red arrowheads point to the center of the pharyngeal terminal bulb. Green arrowheads indicate AMsh cell body position. Scale bar, 10 µm. In d, f, and i, data are represented as mean ± SEM. One-way ANOVA test. **, $p < 0.01$; Each point represents at least 30 worms. In (e and g), data are represented as mean ± SEM. Two-way ANOVA test. **, $p < 0.01$; ns, no significant difference. Each point represents three experiments of at least 50 worms each.

The online version of this article includes the following figure supplement(s) for figure 3:

**Figure supplement 1.** Identification of SAX-3 interacting proteins by proteomics.
**Figure supplement 2.** SYG-1 is expressed in AMsh glia.

some Robo-mutant phenotypes are not related to Slit (*Zallen et al., 1998*; *Hao et al., 2001*; *Fricke et al., 2001*; *Hivert et al., 2002*; *Liu et al., 2004*). In *C. elegans*, there is only one Robo, *sax-3*, and one Slit, *slt-1*. *sax-3(lf)* animals display many phenotypes that are not observed in *slt-1* mutants, including embryonic lethality, abnormal head morphology (*vab* head), and misplacement of neurons (*Zallen et al., 1998*; *Hao et al., 2001*). Based on these observations, it has been long speculated that there are Slit-independent Robo regulatory mechanisms. Here we present evidence to show that neuron-expressed Robo/SAX-3 can be cleaved in the extracellular FN3 repeats during development, and the cleaved extracellular fragment can interact with glial-expressed SYG-1/Neph to mediate F-actin organization and to facilitate glial migration.

The extracellular region of Robo is evolutionarily conserved from nematodes to mammals and contains five Ig and three FN3 domains. The extracellular cleavages of Robo appear to be conserved in *C. elegans* (this study), *Drosophila*, and human (*Coleman et al., 2010*; *Seki et al., 2010*), though different metalloproteases were utilized in *Drosophila* and human. Recent biochemical and structural studies show that the extracellular domains of Robo can dimerize through the Ig domains, and the dimerization may be inhibited by the FN3 domain in the absence of Slit (*Hivert et al., 2002*; *Liu et al., 2004*; *Zakrys et al., 2014*; *Aleksandrova et al., 2018*; *Barak et al., 2019*; *Yamamoto et al., 2019*; *Pak et al., 2020*). The cleavages of the extracellular domains provide

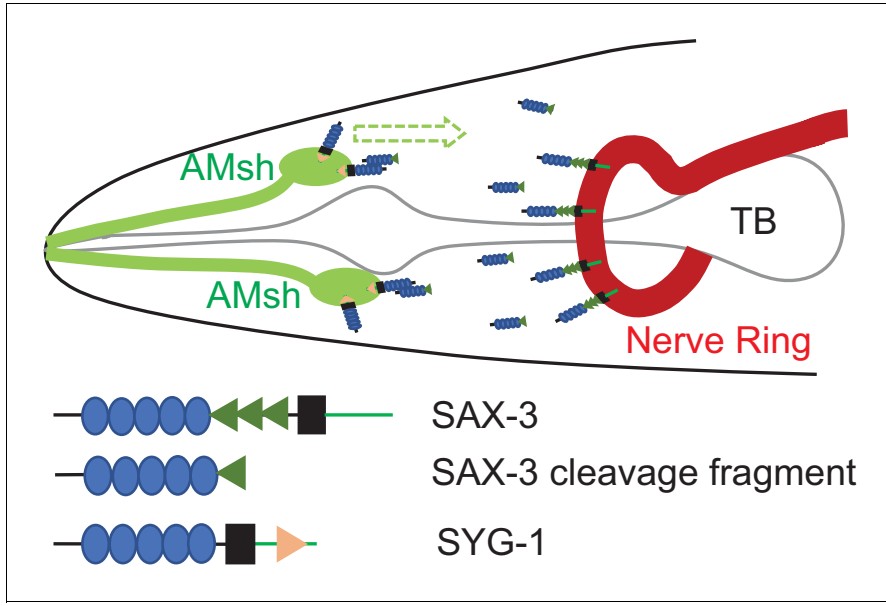

**Figure 4.** A model for regulation of AMsh glial migration by SAX-3 and SYG-1.

possibility for additional regulatory mechanisms of Robo. Since the extracellular SAX-3/Robo cleavage fragment lacks the inhibitory second FN3 domain, it will be expected to strongly bind with itself or with the full length SAX-3/Robo, and these interactions may play a regulatory role in the activation of Robo downstream signals. In this study we show that the SAX-3/Robo extracellular cleavage fragment can bind with the Ig domain-containing adhesion molecule SYG-1/Neph to regulate the WAVE complex. Since *sax-3*/Robo and *syg-1/* Neph are both expressed in and functionally important for neurons and other cells, it is possible that interactions between the SAX-3 cleaved extracellular fragment and *syg-1/*Neph also play functions beyond glial migration. The potential dimerization of SAX-3/Robo extracellular cleavage fragments may also mediate crosstalk between *syg-1*/Neph and *sax-3/* Robo signals. The extracellular cleavage of SAX-3/Robo generated two fragments: one has all the Ig and the first FN3 domains while the other contains the rest of the FN3, transmembrane, and the intracellular domains. While the released SAX-3/Robo extracellular cleavage fragment functions as a ligand for SYG-1/Neph and has potential to interact with the full length SAX-3/Robo, the membrane-anchored fragment may also play an active role in neuronal development.

To achieve their vital roles in the nervous system and to form functional units with neurons, glia need to migrate over what are often long distances from their birthplace to the appropriate regions (*Gilmour et al., 2002*; *Jarjour and Kennedy, 2004*; *Kinrade et al., 2001*; *Klämbt, 2009*; *Barres, 2008*; *Fields et al., 2015*; *Mori et al., 2005*). The migration of glia relies on guidance cues and shares many similar mechanisms with neurons (*Kinrade et al., 2001*; *Jarjour et al., 2003*; *Sasse and Klämbt, 2016*; *Spassky et al., 2002*; *Unni et al., 2012*). Our studies uncover a surprising role of Robo receptors – an attractive cue for glial migration – and identify SYG-1/Neph as the glial receptor for the neuronal-released Robo cleavage fragment, suggesting that the function and regulation of guidance molecules are more complicated than anticipated.

## Materials and methods

Further information and requests for resources should be directed to and will be fulfilled by the Lead Contact, Dong Yan (dong.yan@duke.edu).

### Experimental model and subject details

#### *C. elegans* genetics

*C. elegans* strains were maintained on nematode growth media (NGM) plates using *Escherichia coli* OP50 as a food source. Animals were grown according to standard methods (*Brenner, 1974*) at 20° C unless otherwise stated. Wild-type worms were of the Bristol N2 strain. Only hermaphrodite animals were used for experiments. All transgenes and strains are described in *Supplementary file 1*. *yadIs48* (*Pf53f4.13::GFP*) was used to visualize AMsh cells. The null alleles used in genetic analyses were *sax-3(yad10)*, *slit-1 (ok255)*, *syg-1(ok3640)*, *syg-2(ky671)*, and *gex-3(zu196)* unless specific alleles were mentioned. *gex-3(zu196)* homozygous animals are sterile, and we analyzed the AMsh phenotypes in first generation homozygotes of *gex-3(zu196)*, which may not represent null allele phenotypes due to maternal effects. All other data were acquired in animals that have been homozygous for genotypes of interest for at least three generations. The recessive *sax-3* alleles *yad10* and *yad147* were isolated from a visualized EMS mutagenesis screen of over 4000 haploid genomes and were two of five mutants with reduced migration defects. Genetic studies show that those five mutants belong to three complementation groups, and *yad10* and *yad147* are in one complementation group. Based on their X-linkage and the *vab* head phenotypes, we tested whether they were alleles of *sax-3* and found that they both failed to complement *sax-3(ky123)* null alleles. *yad10* and *yad147* mutations were finally confirmed through sequencing and rescue experiments. The '*vab*' head phenotype in *sax-3* mutants was defined as any abnormality of head morphology.

### Method details

#### Cloning and constructs

All DNA expression constructs were generated using Gateway cloning technology (Invitrogen, Carlsbad, CA) and subsequently sequenced. *sax-3*, *syg-1,* and *gex-3* cDNA were amplified from *C. elegans* yk cDNA clones gifted from Dr. Yuji Kohara's lab. *Pf53f4.*13 (PAMsh), *Pf16f9.3* (PAMsh 2), *Punc-33* (Pneuron), and *Prgef-1*(Pneuron) promoters were used to express genes of interest in AMsh

and all neurons. *Parl-13* was used to drive gene expression in amphid neurons. In embryos and other stages, *f53f4.13* and *f16f9.3* are exclusively expressed in AMsh glia, *arl-13* is exclusively expressed in amphid neurons, and *unc-33* and *rgef-1* are broadly expressed in neurons (*Packer et al., 2019*). A complete list of DNA constructs used is included in *Supplementary file 1*. In general, plasmids used in this study were injected at a concentration of 1–50 ng/μL with a *Pttx-3::RFP* co-injection marker injected at a concentration of 50 ng/μL.

## Microscopy

Representative images were acquired with a Zeiss LSM700 confocal microscope using a Plan-Apochromat 40×/1.4 objective. Worms were immobilized using 1.5% 1-phenoxy-2-propanol (TCI America, Portland, OR) in M9 buffer and mounted on 5% agar slides. 3D reconstructions were done using Zeiss Zen software. A Zeiss Axio Imager two microscope equipped with Chroma HQ filters was used to score AMsh migration defects and take images to measure Migration Index. Experiments were conducted in triplicates consisting of at least 50 day one adult animals each.

## Protein analysis

All protein analyses were carried out using transgenes made in this study (see details in *Supplementary file 1*). Synchronous animals were grown on NGM plates to reach adult stages, and animals were bleached to collect embryos. The freshly isolated embryos were mostly between the 16-cell and 44-cell stages. Embryos were incubated in M9 buffer and collected at different embryonic stages or at the L1 stage. L1 animals were then cultured in NGM plates with O.P. 50 as a food source until they reached the L4 stage, where they were subsequently collected for protein analysis. For western blotting, protein was directly extracted in SDS sample buffer containing 1 mM DTT by freeze-thawing for 20–50 cycles between dry ice/ethanol and a 37°C water bath, and then denatured by heating to 95°C for 5 min. Blots were probed with anti-HA (Sigma, H9658; abcam, ab9110), anti-GFP (Sigma, G1544), and anti-FLAG (Sigma, F3165) antibodies and visualized with Amersham HRP-conjugated anti-rabbit or anti-mouse secondary antibodies at a 1:5000 dilution (GE Healthcare Life Sciences) using the SuperSignal West Femto kit (Pierce, Rockford, IL). For IP experiments, embryos were collected at the 44-cell and 1.5- to 3-fold stages and subsequently lysed with IP lysis buffer (25 mM Tris-HCl pH 7.4, 150 mM NaCl, 1% NP-40, 1 mM EDTA, 5% glycerol) containing protease inhibitor cocktail (Thermo Fisher, 87785, added before use) and sonicated by an ultrasonic cell crusher. Lysates were centrifuged for 10 min at 12,000 rpm at 4°C and supernatant was used for Co-IP using the anti-Protein A/G magnetic beads (Thermo Fisher, 78608) or anti-HA Magnetic beads (Thermo Fisher, 88836) separately according the manufacturer's protocol. Briefly, beads were equilibrated before use and added to the supernatant and mixed gently. The mixture was incubated on a rotator at 4°C overnight. Beads were collected with a magnetic stand and washed twice with phosphate buffered saline. Supernatant was discarded and proteins were eluted from the beads using SDS-sample buffer at 95°C for 10 min.

## Collection of supernatant for western blot

### RT-PCR

To measure *sax-3* mRNA levels in N2, *yad10*, and *yad147* animals, total RNA were isolated from mixed-stage animals. Briefly, worms were collected in the tube and washed with M9 buffer on an end-over-end rotator three times for 90 min in total. RNA was then isolated by using TRI Reagent (Invitrogen). Total RNA (1 μg) of each genotype was reverse transcribed using an iScript Reverse Transcription Supermix (Bio-Rad Laboratories). Relative *sax-3* mRNA levels were determined by PCR with 1 μL cDNAs as templates, and *act-1/actin* was used as internal control. Primers for *sax-3*, Forward: 5′ CTGTTTGATTGTCGTGTGACTGG-3′; Reverse: 5′-AAAGCTGCCTTGCTCAACG-3′. Primers for *act-1*, Forward: 5′-TTGCCGCTCTTGTTGTAGAC-3′; Reverse: 5′-TTCGTAGATTGGGACGGTG-3′. The PCR products were run in the agarose gel, and the density was calculated using Image J software.

### CRISPR-Cas9-based homology recombination approach

The *sax-3* FN3-b deletion allele, *yad175,* was constructed by using CRISPR-Cas9-based homology recombination. Briefly, two arms containing about 1000 bp on both sides of FN3-b of *sax-3* were

amplified from genomic DNA of wild-type animals, and the fusion fragment of the two arms was then cloned into the PCR8 vector. Two sgRNAs of *sax-3* FN3-b deletion were cloned into pDD122 plasmid (Addgene, #47550). Finally, the recombination template and the two sgRNAs were injected into young adult of *yadls48* worms simultaneously. *yad175* were isolated from the F2 generation, and the deletion was confirmed by sequencing. *sax-3* sgRNA1: GGATGGAGAGTCAACATG; sequence of *sax-3* sgRNA2: GGATGTGCGAATCCGTAT.

## Mass spectrometric analysis

To identify the potential proteins binding with SAX-3, embryos from FLAG::SAX-3::GFP expressing worms were collected at the 44-cell and 1.5- to 3-fold stages. In the same experiment, 1.5- to 3-fold stage embryos of FLAG::INX-3::GFP transgenes were also collected as a control to further preclude unspecific proteins. Next, embryos were subsequently lysed with IP lysis buffer (25 mM Tris-HCl pH 7.4, 150 mM NaCl, 1% NP-40, 1 mM EDTA, 5% glycerol) containing protease inhibitor cocktail (Thermo Fisher, 87785, added before use) and sonicated by an ultrasonic cell crusher. Lysates were centrifuged for 10 min at 12,000 rpm at 4°C and supernatant was used for Co-IP to pull down all the proteins that bind to SAX-3 by using the anti-FLAG magnetic agarose beads (Thermo Fisher, A36798) according to the manufacturer's protocol.

IP products were sent to the Duke Center for Genomic and Computational Biology for mass spectrum analysis. Briefly, these in-solution samples were brought to 4% SDS and then subjected to S-trap (Protifi) trypsin digestion and sample cleanup using manufacturer recommended protocols. Digested peptides were lyophilized to dryness and resuspended in 15 µL of 0.2% formic acid/2% acetonitrile. Each sample was subjected to chromatographic separation on a Waters NanoAquity UPLC equipped with a 1.7 µm HSS T3 C18 75 µm I.D. $\times$ 250 mm reversed-phase column (NanoFlow data). The mobile phase consisted of (A) 0.1% formic acid in water and (B) 0.1% formic acid in acetonitrile. Of this, 3 µL was injected and peptides were trapped for 3 min on a 5 µm Symmetry C18 180 µm I.D. $\times$ 20 mm column at 5 µL/min in 99.9% A. The analytical column was then switched in-line and a linear elution gradient of 5% B to 40% B was performed over 30 min at 400 nL/min for in-gel band analysis and over 90 min at 400 nL/min for Co-IP studies. The analytical column was connected to a Fusion Lumos mass spectrometer (Thermo) through an electrospray interface operating in a data-dependent mode of acquisition. The instrument was set to acquire a precursor MS scan from m/z 375–1500 at R = 120,000 (target AGC 2e5, max IT 50 ms) with MS/MS spectra acquired in the ion trap (target AGC 5e3, max IT 100 ms). For all experiments, HCD energy settings were 30 v and a 20 s dynamic exclusion was employed for previously fragmented precursor ions.

Raw LC-MS/MS data files were processed in Proteome Discoverer (Thermo Scientific) and then submitted to independent Mascot searches (Matrix Science) against a *C. elegans* protein database containing both forward (4043 entries) and reverse entries of each protein. Search tolerances were five ppm for precursor ions and 0.8 Da for product ions using trypsin specificity with up to two missed cleavages. Carbamidomethylation (+57.0214 Da on C) was set as a fixed modification, whereas oxidation (+15.9949 Da on M) and deamidation (+0.98 Da on QN) were considered dynamic mass modifications. All searched spectra were imported into Scaffold (v4.4, Proteome Software) and scoring thresholds were set to achieve a peptide false discovery rate of 1% using the PeptideProphet algorithm.

## Quantification and statistical analyses

### Statistical analysis

Data were analyzed using Student's t-test and one-way or two-way ANOVA followed by Tukey's post-hoc test in Graphpad Prism (Graphpad Software, La Jolla, CA). Statistical details are included in the figure legends. In general, a p-value cutoff of 0.05 was considered statistically significant. The data are represented by mean ± SEM. For phenotype penetrance experiments, three biological replicates of at least 50 animals each were used unless otherwise stated.

## Acknowledgements

We thank Dr. Erik Soderblom, Tricia C Ho and Dr. Greg Waitt at Duke Proteomics and Metabolomics Core Facility for helping on sample preparation, mass spectrum analyses, and data interpretation,

and Dr. Yuji Kohara for *syg-1*, *sax-3*, and *gex-3* cDNAs. Some strains used in this study were provided by the Caenorhabditis Genetics Center (CGC), which is funded by the NIH Office of Research Infrastructure Programs (P40 OD010440). Ms. Chia-hui Chen characterized some preliminary observations of this study. This project was supported by the Holland Trice Awards. QZ, AZ, and DY were supported by the NIH R01 (NS094171 and NS105638 to DY).

## Additional information

### Funding

| Funder | Grant reference number | Author |
|---|---|---|
| National Institute of Neurological Disorders and Stroke | NS094171 | Dong Yan |
| National Institute of Neurological Disorders and Stroke | NS105638 | Dong Yan |
| School of Medicine, Duke University | The Holland Trice Awards | Dong Yan |

The funders had no role in study design, data collection and interpretation, or the decision to submit the work for publication.

### Author contributions

Zhongwei Qu, Data curation, Formal analysis, Validation, Investigation, Methodology, Writing - original draft, Writing - review and editing; Albert Zhang, Data curation, Formal analysis, Validation, Investigation, Visualization, Methodology, Writing - original draft, Writing - review and editing; Dong Yan, Conceptualization, Supervision, Funding acquisition, Validation, Writing - original draft, Writing - review and editing

### Author ORCIDs

Zhongwei Qu (iD) https://orcid.org/0000-0002-6994-0530
Albert Zhang (iD) https://orcid.org/0000-0003-1310-5680
Dong Yan (iD) https://orcid.org/0000-0002-7542-1251

### Decision letter and Author response

Decision letter https://doi.org/10.7554/eLife.57921.sa1
Author response https://doi.org/10.7554/eLife.57921.sa2

## Additional files

### Supplementary files

• Supplementary file 1. Strains and plasmids used in this study.

• Supplementary file 2. A list of protein identified by proteomics show specific interactions with SAX-3 in 1.5- to 3-fold stage embryos.

• Transparent reporting form

### Data availability

All data generated or analysed during this study are included in the manuscript and supporting files.

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
