## [Decision Letter]

**Acceptance summary:**

Your manuscript identified a new function of the SAX-3/ROBO guidance receptor in regulating glial cell migration. The finding that the ROBO ectodomain is secreted and interacts with a Ig adhesion receptor to guide glial cell migration broadens our understanding of receptor-ligand interactions in developmental neurobiology.

**Decision letter after peer review:**

Thank you for submitting your article "Robo functions as an attractive cue for glial migration through SYG-1/Neph" for consideration by *eLife*. Your article has been reviewed by three peer reviewers, including Kang Shen as the Reviewing Editor and Reviewer #1, and the evaluation has been overseen by Jonathan Cooper as the Senior Editor.

The reviewers have discussed the reviews with one another and the Reviewing Editor has drafted this decision to help you prepare a revised submission.

Summary:

All three reviewers found that this work is interesting for several reasons. For example, one reviewers wrote "Dong and colleagues described a slit independent function of the well-known axon guidance receptor ROBO/SAX-3 in the migration of glial cells in *C. elegans*. They found that both ROBO and an IgSF protein SYG-1 were required for the glial cell migration. Through structure-function analyses, they found the extracellular domain of ROBO was sufficient to rescue the phenotype when overexpressed and they proposed that the extracellular domain of ROBO functions as a secreted ligand to attract SYG-1 expressing glial cell migration. The novelty of this manuscript lies in two folds. First, the development of glial cells are poorly understood. Second that this is the first paper to assign a secreted function of ROBO through interaction with SYG-1. The majority of the genetic data supports each other and is consistent with the model they proposed."

Essential revisions:

However, all three reviewers found that there are important experiments that need to be performed in order to fully support the model. They mostly fall into three areas. First, the use of mis-expression constructs using exogenous promoters is potentially risky in drawing conclusions about cellular locus of SAX-3 in this case. Second, inclusion of cleaner genetic experiments such as a knockin will strengthen the model. Three, there are several concerns about the masspec and biochemistry data presented. The detailed comments are as following:

1) Cell autonomy

Misexpression of SAX-3 and SYG-1 high-copy transgenes under control of Punc-33 for neurons and Pf53f4.13 for the amphid sheath cell is used to infer cell autonomy. There are three issues with this – first, these promoters are not specific, in particular Punc-33 is expressed prominently in the amphid socket which could also affect amphid sheath migration (Altun-Gultekin et al. Development 2001). Pitr-1 used for the over-migration experiment is also expressed in the amphid socket. Second, promoters can be expressed differently in embryos than larvae, so images of the expression pattern of Pf53f4.13 in 1.5 fold embryos in particular should be provided. Third, it is risky to base such a major claim on misexpression experiments especially since promoters can be expressed transiently in unexpected places, so these experiments should really include mosaic analysis as well. Alternatively, the authors can use additional sets of promoters to strengthen the arguments. For example, an amphid neuron promoter might be able to tested, as well as neurons that do not associate with the glial cells but still has a process in the nerve ring such as PVQ or AVA. There are promoters available for those experiments.

Figure 3H and I and crucial for the model. These are quite compelling results but I am not entirely sure about the interpretation of these results. The DA9 expression of itr-1 promoter does not turn on consistently until L3-L4. Do the sheath cells continue to grow in the *sax-3* single mutants expressing the transgene? To mimic more endogenous situation, the authors should use the lin-44 promoter which expresses embryonically.

2) Overexpression issues.

All rescue experiments are based on overexpression. At least one knockin should be done to show that the FN3(2) is required. For example, if FN3(2) is deleted using crispr, they can try to test if the it is required for slit function and for this new function of *sax-3*.

3) Binding to SYG-1

The syg-1 phenotype is convincing but the SAX-3 binding data is less convincing. First, the list of hits from the mass spec screen is not shown. How many other hits were there, what were they, and how did the representation of syg-1 compare? Second, there are no specificity controls (for the antibody or for the SAX-3-SYG-1 binding itself) in the IP experiments. This is especially important because the SAX-3 fragment is running at the wrong size in this experiment compared to all other gels in the paper.

Related to this issue, A. The mass spectrometric analysis is not explained. Presumably, the authors identified hundreds of peptides, many of which might have been non-specific – a common issue with such analysis. The fact that only two peptides covering 5% of SYG-1 were identified by mass spec (buried in supplement and not mentioned in text) implies that the confidence level for this identification was likely not high. This is generally fine; the authors followed up this finding with additional experiments. But a proper report of total number of peptides identified, cutoffs, and how SYG-1 stands out from the list of peptides is warranted. I suggest authors contact their mass spec facility for details and statistical analysis.

Additionally, the supplement shows one Asn labeled green in the second peptide by the mass spec analysis software. Does that imply a chemical modification/artifact that affected the mass of the peptide? The supplement, which is essential, could be explained better. Currently, it is a screen snapshot from an unknown piece of software that is not mentioned or cited.

B) The authors should report exact boundaries of their deletion constructs (ΔC, ΔC+TM, Ig(1-5)FN3(1), etc.) presented in Figures 1, 2 and 3, especially since they would be relevant for judging whether their manipulations (domain deletions or inclusions) could inadvertently cause misfolding of proteins, or where exactly the cleavage might be happening. It would help narrow down a binding site on the sequence for the proposed SYG-1 interaction. In addition, we'd know if they removed the long linker sequences shown to be important between some of the IG and FN domains (Barak et al., 2019; Aleksandrova et al., 2018; Pak et al., 2020), when they removed the domains. It would be of interest to see if proteolysis reported by Qu et al. matches that observed by Pak et al., 2020, between the first and second FN domains of human Robo3. On that remark, I cannot find how they draw their schematics for Figure 4 degradation product (containing two FN domains).

C) Authors present glial migration phenotypes for a set of truncations and deletions of SAX-3. The data provided is self-consistent with the biochemistry they present (which is sparse but enough for an initial study of this kind). However, they do not present data that shows that these deletions (such as IG(1-5) or FN3(1) in Figure 2, which did not rescue) are successfully expressed, make it to through the secretory machinery and are presented on/secreted from neurons. One exception is the FN3(2)Δ constructs, where they did show protein production, but not necessarily successful display on cells. These are useful controls that actually allow them to support their conclusions. It would also be preferable to see if the expression levels of these deletions are comparable to wild-type, so that the observed phenotypes are not due to underexpression of or overproduction of SAX-3. Due to the current circumstances with the coronavirus pandemic, the authors may be given a dispensation from completing these experiments, as long as they clarify the possibility that some of these constructs might not be produced and secreted/displayed in vivo, and hence the lack of rescue.

D) I do not understand the comment about FN domains being mostly structural, while IG domains are functional. If this is a general comment about IG/FN proteins such as Robo, one of the most prominent IG/FN receptors DCC/UNC-40 binds its Netrin ligand with the FN domains, not IG. If this is a comment about the FN domains of Robos not being "functional", that's also not appropriate: While Slits bind the Robo IG domain 1, the other known extracellular ligand of Robos, the NELL proteins, bind the first FN domain of vertebrate Robos (Pak et al., 2020 and Yamamoto et al., 2019), the same domain identified by the authors for SYG-1 binding, and NELLs don't touch the IG domains.

In fact, the authors may want to discuss the relevance of the FN domains for Robo function, which was recently resolved in a burst of publications (the Pak et al., Yamamoto, et al., Barak et al. and Aleksandrova et al. manuscripts). The cleavage described here might actually act to open Robo receptors for binding other ligands, such as NELLs (not present in *C. elegans*) or SYG-1.

E).The authors do not show that the interaction is direct at any point in the manuscript. The pull-downs can simply be the result of a third protein mediating this interaction; this is not an uncommon occurrence. This does not make the paper less interesting, but the possibility has to be clearly mentioned across the manuscript.

[Editors' note: further revisions were suggested prior to acceptance, as described below.]

Thank you for resubmitting your work entitled "Robo functions as an attractive cue for glial migration through SYG-1/Neph" for further consideration by *eLife*. Your revised article has been evaluated by Jonathan Cooper (Senior Editor) and Kang Shen as a Reviewing Editor.

The manuscript has been improved and but there are some remaining issues that need to be addressed before acceptance, as outlined below: Can you address these points by providing more information and clarification in your writing when providing the revision.

Reviewer #2:

1) Cell autonomy experiments still depend exclusively on promoters whose expression and timing in the embryo, relative to AMsh migration, have not been determined. The images of AMsh-specific promoters (Figure 1—figure supplement 1G) appear to show different cells, one much more anterior than the other. Mosaic analysis was not performed.

2) In the mass spectrometry experiments, it appears that SYG-1 was one of the weakest hits detected, ranking below 1000 in the overall list. Even with aggressive filtering, SYG-1 is among a few dozen hits and appears at the level of background noise, consistent with the presence of a "DECOY" hit with identical results. This is not the impression given in the manuscript.

3) The relative position of the nerve ring and the AMsh in Figure —figure supplement 1D strongly suggests that the AMsh is tracking with the displaced nerve ring, contrary to the authors' interpretation. Due to COVID, the authors could not obtain videos of glial migration.

4) *yad10* is expected to encode a protein similar to the Ig(1-5)FN(1-3) fragment used for rescue. One possibility was that the mRNA is not expressed but the authors now show that is not the case. The *yad10* phenotype contradicts the overexpression results in which the N-terminal fragment of SAX-3 is sufficient for rescue.

Reviewer #3:

The authors responded to comments satisfactorily as much as I can judge. My geneticist colleagues should review the validity of the new experiments with the new promoters and the CRISPR deletion of the second FN3 domain.

I am not sure the statement "Recent biochemical and structural studies show that the extracellular domains of Robo can dimerize through the fourth Ig domain, and this dimerization is inhibited by the second FN3 domain in the absent of Slit" is strictly true. It may be safest to define this as "FN3 domains", since both are involved FN3 domains are involved in a hairpin-like structure. Also, there is plenty of evidence that Robo1 is a dimer even when the FN3 domains are there, so I would say "the dimerization may be inhibited" to allow for multiple possibilities. Also see the typo, "in the absent of".

Also, the last statement appears to be false. NELLs bind the first FN3 domain, not second, so they would not be binding the membrane-anchored fragment. NELLs probably should not be mentioned here at the end of the text at all. My initial comment was that the first FN3(1) domain is not a non-functional/only structural domain as implied by the original draft. NELLs do bind the first FN3 domain, and can turn on a repulsive Robo response.

I should also remind the authors that they still use inconsistent naming of the Fibronectin domains. Figure 1—figure supplement 1 has FN3-a and FN1 for the same domains in panels F and H. Or there is FN1 in Figure 3I, but refers to this domain as FN3-a in Figure 2F. Minor issue, but no reason to confuse readers.

---

## [Author Response]

Essential revisions:However, all three reviewers found that there are important experiments that need to be performed in order to fully support the model. They mostly fall into three areas. First, the use of mis-expression constructs using exogenous promoters is potentially risky in drawing conclusions about cellular locus of SAX-3 in this case. Second, inclusion of cleaner genetic experiments such as a knockin will strengthen the model. Three, there are several concerns about the masspec and biochemistry data presented. The detailed comments are as following:1) Cell autonomyMisexpression of SAX-3 and SYG-1 high-copy transgenes under control of Punc-33 for neurons and Pf53f4.13 for the amphid sheath cell is used to infer cell autonomy. There are three issues with this – first, these promoters are not specific, in particular Punc-33 is expressed prominently in the amphid socket which could also affect amphid sheath migration (Altun-Gultekin et al. Development 2001). Pitr-1 used for the over-migration experiment is also expressed in the amphid socket. Second, promoters can be expressed differently in embryos than larvae, so images of the expression pattern of Pf53f4.13 in 1.5 fold embryos in particular should be provided. Third, it is risky to base such a major claim on misexpression experiments especially since promoters can be expressed transiently in unexpected places, so these experiments should really include mosaic analysis as well. Alternatively, the authors can use additional sets of promoters to strengthen the arguments. For example, an amphid neuron promoter might be able to tested, as well as neurons that do not associate with the glial cells but still has a process in the nerve ring such as PVQ or AVA. There are promoters available for those experiments.Figure 3H and I and crucial for the model. These are quite compelling results but I am not entirely sure about the interpretation of these results. The DA9 expression of itr-1 promoter does not turn on consistently until L3-L4. Do the sheath cells continue to grow in the sax-3 single mutants expressing the transgene? To mimic more endogenous situation, the authors should use the lin-44 promoter which expresses embryonically.

In the revised manuscript we repeated the key rescue experiments using a well established Pneuronal promoter, Prgef-1, an amphid neuron specific promoter, Parl-13, and another AMsh specific promoter Pf16f9.3, and all results supported our original conclusions that *sax-3* and syg-1 function in neurons and AMsh, respectively, to regulate AMsh migration (Figure 1D, Figure 2—figure supplement 1D, and Figure 3—figure supplement 1B). We also provided new data to show that both Pf53f4.13 (PAMsh) and Pf16f9.3 (PAMsh(2)) can drive expression in AMsh glia of 1.5-fold stage embryos (Figure 1—figure supplement 1G). To further confirm that ectopic expression of the SAX-3 extracellular cleavage fragment can attract AMsh migration in a syg-1 dependent manner, we expressed SAX-3(Ig1-5-FN3a) under the Plin-44 promoter, as suggested by the reviewer, and found that AMsh glial displayed the same syg-1 dependent AMsh migration defects as that in Pitr-1 transgenes (Figure 3H and 3I).

2) Overexpression issues.All rescue experiments are based on overexpression. At least one knockin should be done to show that the FN3(2) is required. For example, if FN3(2) is deleted using crispr, they can try to test if the it is required for slit function and for this new function of sax-3.

In the revised manuscript, we generated a deletion allele of *sax-3*(*yad175*) by CRISPR-mediated recombination. *yad175* has an in frame deletion that removes the coding sequence of the second repeat of FN3 (FN3-b). *yad175* displays much stronger AMsh migration defects than *sax-3*(ky123, suggested null allele) and *sax-3*(*yad10*), while having relatively weaker vab head and AVM guidance defects than those in ky123 and *yad10* animals (Figure 1—figure supplement 1F, and Figure 2—figure supplement 1E). The vab head and AVM guidance defects are likely associated with the low *sax-3* mRNA level in *yad175* (about 50% of that in control animals, Figure 2—figure supplement 1D), which is likely caused by the deletion itself. In summary, even though being a weak allele, *sax-3*(*yad175*) has stronger defects in AMsh migration, further arguing the importance of the second FN3 repeat/SAX-3 cleavage in AMsh migration. The stronger AMsh migration defect displayed in *yad175* also suggests that ky123 may not be a completely null allele of *sax-3*.

3) Binding to SYG-1The syg-1 phenotype is convincing but the SAX-3 binding data is less convincing. First, the list of hits from the mass spec screen is not shown. How many other hits were there, what were they, and how did the representation of syg-1 compare? Second, there are no specificity controls (for the antibody or for the SAX-3-SYG-1 binding itself) in the IP experiments. This is especially important because the SAX-3 fragment is running at the wrong size in this experiment compared to all other gels in the paper.Related to this issue, A. The mass spectrometric analysis is not explained. Presumably, the authors identified hundreds of peptides, many of which might have been non-specific – a common issue with such analysis. The fact that only two peptides covering 5% of SYG-1 were identified by mass spec (buried in supplement and not mentioned in text) implies that the confidence level for this identification was likely not high. This is generally fine; the authors followed up this finding with additional experiments. But a proper report of total number of peptides identified, cutoffs, and how SYG-1 stands out from the list of peptides is warranted. I suggest authors contact their mass spec facility for details and statistical analysis.

In the revised manuscript, we provided a list of all SAX-3 binding proteins that were identified only in the 1.5-3 fold stage embryos (Supplementary file 2). In this experiment, we compared the proteins identified from IP-Mass spec of FLAG::SAX-3 in embryos in 16-44-cell and 1.5-3 fold stages, and identified proteins with specific interactions with SAX-3 only in 1.5-3 fold stage embryos. We also ran a similar experiment using IP of FLAG::INX-3(innexin/gap junction) in 1.5-3 fold stage embryos as a negative control for FLAG IP. All peptides identified were beyond cutoff and have a high confidence as mentioned in the new Materials and methods section. We have also added the detailed description of proteomics and data analyses in the updated Materials and methods section.

In Figure 2, the HA tag was fused at the end of SAX-3 C-terminal, and the “cleavage” band detected by the anti-HA antibody is the leftover fragment of the extracellular cleavage (about 70 kDa). In Figure 3, the FLAG tag was fused at the beginning of SAX-3 N-terminal (right after the signal peptide), and the cleavage band detected by the anti-FLAG antibody is the extracellular cleavage product (about 50-55 kDa). We apologize for the confusion, and have highlighted the position of HA and FLAG tags and the predicted size of each band in the updated figures. We also included additional controls (IgG control and transgene control) in the updated Figure 3B, and ran the IP products and inputs of SAX-3 in the same gel in which the SYG-1-Co-IPed SAX-3 extracellular fragment is about 55 kDa.

Additionally, the supplement shows one Asn labeled green in the second peptide by the mass spec analysis software. Does that imply a chemical modification/artifact that affected the mass of the peptide? The supplement, which is essential, could be explained better. Currently, it is a screen snapshot from an unknown piece of software that is not mentioned or cited.

The screen shot was the mass spec results analyzed by software called “Scaffold”. The green color of the Asparagine(N) means this amino acid was deamidated to either aspartic acid or isoaspartic acid. We added those details in the updated figure legend and Materials and methods sections. Thanks!

B) The authors should report exact boundaries of their deletion constructs (ΔC, ΔC+TM, Ig(1-5)FN3(1), etc.) presented in Figures 1, 2 and 3, especially since they would be relevant for judging whether their manipulations (domain deletions or inclusions) could inadvertently cause misfolding of proteins, or where exactly the cleavage might be happening. It would help narrow down a binding site on the sequence for the proposed SYG-1 interaction. In addition, we'd know if they removed the long linker sequences shown to be important between some of the IG and FN domains (Barak et al., 2019; Aleksandrova et al., 2018; Pak et al., 2020), when they removed the domains. It would be of interest to see if proteolysis reported by Qu et al. matches that observed by Pak et al., 2020, between the first and second FN domains of human Robo3. On that remark, I cannot find how they draw their schematics for Figure 4 degradation product (containing two FN domains).

In the revised manuscript, we added a description of exact boundaries of their deletion constructs in Figure 1—figure supplement 1H. The linker sequences were maintained in the deletion constructs. We revised the model and the SAX-3 extracellular cleavage fragments now only contain the first FN3 repeat. We apologize for the oversight.

C) Authors present glial migration phenotypes for a set of truncations and deletions of SAX-3. The data provided is self-consistent with the biochemistry they present (which is sparse but enough for an initial study of this kind). However, they do not present data that shows that these deletions (such as IG(1-5) or FN3(1) in Figure 2, which did not rescue) are successfully expressed, make it to through the secretory machinery and are presented on/secreted from neurons. One exception is the FN3(2)Δ constructs, where they did show protein production, but not necessarily successful display on cells. These are useful controls that actually allow them to support their conclusions. It would also be preferable to see if the expression levels of these deletions are comparable to wild-type, so that the observed phenotypes are not due to underexpression of or overproduction of SAX-3. Due to the current circumstances with the coronavirus pandemic, the authors may be given a dispensation from completing these experiments, as long as they clarify the possibility that some of these constructs might not be produced and secreted/displayed in vivo, and hence the lack of rescue.

In the revised manuscript, we provided new data to show that SAX-3(ΔFN3-b)::GFP has similar expression pattern and level as those of the full length of SAX-3::GFP (Figure 2—figure supplement 1C). We also examined the expression of IG(1-5)::GFP and FN1::GFP. However, these fragments are secreted into extracellular space and have very different expression patterns when compared with the full length of SAX-3::GFP, and we were not able to compare their expression with SAX-3 full-length. Therefore we toned down our conclusions regarding IG(1-5) and FN3(1). Thanks!

D) I do not understand the comment about FN domains being mostly structural, while IG domains are functional. If this is a general comment about IG/FN proteins such as Robo, one of the most prominent IG/FN receptors DCC/UNC-40 binds its Netrin ligand with the FN domains, not IG. If this is a comment about the FN domains of Robos not being "functional", that's also not appropriate: While Slits bind the Robo IG domain 1, the other known extracellular ligand of Robos, the NELL proteins, bind the first FN domain of vertebrate Robos (Pak et al., 2020 and Yamamoto et al., 2019), the same domain identified by the authors for SYG-1 binding, and NELLs don't touch the IG domains.

We apologized for the confusion and have removed the comment about the FN domains in the revised manuscript.

In fact, the authors may want to discuss the relevance of the FN domains for Robo function, which was recently resolved in a burst of publications (the Pak et al., Yamamoto, et al., Barak et al. and Aleksandrova et al. manuscripts). The cleavage described here might actually act to open Robo receptors for binding other ligands, such as NELLs (not present in C. elegans) or SYG-1.

We have revised the Discussion section and highlighted the possibilities of other regulations mediated by the SAX-3 cleavage fragment including NElls. Thanks!

E) The authors do not show that the interaction is direct at any point in the manuscript. The pull-downs can simply be the result of a third protein mediating this interaction; this is not an uncommon occurrence. This does not make the paper less interesting, but the possibility has to be clearly mentioned across the manuscript.

We have updated the manuscript and added a sentence to summarize the IP results as “suggesting that SYG-1 can directly or indirectly bind with the SAX-3 extracellular cleavage fragment”. Thanks

[Editors' note: further revisions were suggested prior to acceptance, as described below.]

The manuscript has been improved and but there are some remaining issues that need to be addressed before acceptance, as outlined below: Can you address these points by providing more information and clarification in your writing when providing the revision.Reviewer #2:1) Cell autonomy experiments still depend exclusively on promoters whose expression and timing in the embryo, relative to AMsh migration, have not been determined. The images of AMsh-specific promoters (Figure 1—figure supplement 1G) appear to show different cells, one much more anterior than the other. Mosaic analysis was not performed.

As our data shown in Figure 1—figure supplement 1G, both P*f53f4.13*, and P*f16f9.3* can drive expression of GFP in AMsh glia in embryos, which is consistent with a recent study published in Science ( “A lineage-resolved molecular atlas of *C. elegans* embryogenesis at single-cell resolution”), in which they showed that both *f53f4.13*, and *f16f9.3* mRNA were exclusively expressed in AMsh glia in embryos. In the same manuscript, they also presented data to show that in embryos *arl-13* was exclusively expressed in amphid neurons, and *unc-33* was broadly expressed in neurons. Based on this evidence, we believe that it is appropriate to reach our conclusion using these promoters, and we have cited the Science manuscript in the Materials and methods section to support the use of these promoters. The pictures of P*f53f4.13* and P*f16f9.3* reporters in embryos are not at identical developmental stages, which causes the appearance of slightly different positions of AMsh glia.

2) In the mass spectrometry experiments, it appears that SYG-1 was one of the weakest hits detected, ranking below 1000 in the overall list. Even with aggressive filtering, SYG-1 is among a few dozen hits and appears at the level of background noise, consistent with the presence of a "DECOY" hit with identical results. This is not the impression given in the manuscript.

We agree with the reviewer that SYG-1 was not abundantly detected in our proteomic studies, but as described in the Materials and methods section their interactions were true signals and were beyond the level of “background noise”. In this experiment, we had two control groups: one was the same transgene at the “non-cleavage” stage, and the other one was a membrane channel protein INX-3 at the same developmental stage, and we didn’t detect any SYG-1 peptides in those groups. The proteomic experiments were done twice, and the detection of SYG-1 peptides in 1.5-3 fold stage embryos but not in control groups were observed both times.

3) The relative position of the nerve ring and the AMsh in Figure 1—figure supplement 1D strongly suggests that the AMsh is tracking with the displaced nerve ring, contrary to the authors' interpretation. Due to COVID, the authors could not obtain videos of glial migration.

In the control animals AMsh glia migrate from their birth place, the nose region of the animals, toward the nerve ring, and terminate their migration at the region just past the nerve ring. There were no control adult animals we observed with any AMsh glia cell body anterior to the nerve ring. If the “AMsh is tracking with the displaced nerve ring”, we would not expect to see any change in the relative position of AMsh glia and the nerve ring in *sax-3(lf)* animals. However, that is not case as shown in Figure 1—figure supplement 1D. In *sax-3(lf)* animals about 60% of adult animals had at least one AMsh glia presiding at a region anterior to the nerve ring and closer to their birth place, supporting our conclusion that the AMsh migration defects are not a secondary effect of the displaced nerve ring. We apologize for the possible confusion, and we have revised this part in the manuscript to make it clearer.

4) yad10 is expected to encode a protein similar to the Ig(1-5)FN(1-3) fragment used for rescue. One possibility was that the mRNA is not expressed but the authors now show that is not the case. The yad10 phenotype contradicts the overexpression results in which the N-terminal fragment of SAX-3 is sufficient for rescue.

How a mutation affects gene expression is complicated, and detectable *sax-3* mRNA does not mean the protein can be expressed, folded, and released in a correct manner. It is still to be determined that how *yad10* affects *sax-3,* but the case like *yad10* is not unusual (at least in *C. elegans*).

Reviewer #3:The authors responded to comments satisfactorily as much as I can judge. My geneticist colleagues should review the validity of the new experiments with the new promoters and the CRISPR deletion of the second FN3 domain.I am not sure the statement "Recent biochemical and structural studies show that the extracellular domains of Robo can dimerize through the fourth Ig domain, and this dimerization is inhibited by the second FN3 domain in the absent of Slit" is strictly true. It may be safest to define this as "FN3 domains", since both are involved FN3 domains are involved in a hairpin-like structure. Also, there is plenty of evidence that Robo1 is a dimer even when the FN3 domains are there, so I would say "the dimerization may be inhibited" to allow for multiple possibilities. Also see the typo, "in the absent of".

Corrected.

Also, the last statement appears to be false. NELLs bind the first FN3 domain, not second, so they would not be binding the membrane-anchored fragment. NELLs probably should not be mentioned here at the end of the text at all. My initial comment was that the first FN3(1) domain is not a non-functional/only structural domain as implied by the original draft. NELLs do bind the first FN3 domain, and can turn on a repulsive Robo response.

We have removed the description about the potential interactions between NELLs and the membrane cleavage product.

I should also remind the authors that they still use inconsistent naming of the Fibronectin domains. Figure 1—figure supplement 1 has FN3-a and FN1 for the same domains in panels f and h. Or there is FN1 in Figure 3I, but refers to this domain as FN3-a in Figure 2F. Minor issue, but no reason to confuse readers.

Revised to FN3-a and FN3-b in all figures.